# Taylor's law predicts unprecedented pulses of forest disturbance under global change

Cornelius Senf ®[1] ✉, Rupert Seidl ®[1,2], Thomas Knoke[1] & Tommaso Jucker ®[3]

Climate extremes are causing increasingly large pulses of forest disturbance across biomes, raising concerns that forests are pushed beyond their safe operating space. However, predicting future disturbance pulses remains a major challenge, as these events are stochastic and driven by complex ecological and socio-economic processes. Here, we provide a tractable solution to this problem using Taylor's law, which predicts changes in variability (and thus the frequency of extremes) from changes in the mean. We empirically test the hypothesis that forest disturbance dynamics can be described through Taylor's law using high-resolution annual disturbance maps of Europe's forests going back 35 years. We find strong evidence for a power law relationship between mean disturbance rates and their temporal variability, indicating that increasing mean disturbance rates – as observed for Europe and many other parts of the globe – significantly amplify the probability of large disturbance pulses. The power law relationship was consistent across natural disturbance agents, spatial grains, and biomes, and applied also to human-driven disturbances. Our findings challenge the assumption that extreme disturbance pulses are inherently unpredictable, providing a data-driven framework for their integration into forest policy and management.

Natural forest disturbances caused by bark beetle outbreaks, windthrow or fire are integral drivers of forest dynamics[1]. They increase landscape heterogeneity, with positive effects on forest biodiversity[2,3] and resilience[4]. However, natural disturbances are among the most climate-sensitive processes in forest ecosystems. Large disturbances pulses are often closely linked to the occurrence of climate extremes, including storms[5-7], droughts[8-10] and heatwaves[11-13], all of which are expected to increase under climate change[14-17]. Pulses of natural disturbances are therefore also expected to increase in the future[10,18,19], but predicting their frequency or severity remains a major challenge with current data and models[20].

Many ecological systems show a power law relationship between their mean and their variance. This pattern was first described by Taylor in 1961[21] when observing temporal fluctuations in population density of different organisms and has since become known as "Taylor's law". Taylor's law holds true for a wide variety of species, including viruses, microorganisms, vertebrates and plants, as well as for many other ecological processes[22]. It has also been applied in other fields beyond ecology[23,24], such as predicting temporal dynamics of infectious diseases like COVID19[25]. Given the apparently universal nature of Taylor's law[26], it seems reasonable to assume that it would also apply to the temporal dynamics of forest disturbances. If true, this would imply that even moderate increases in mean disturbance rates would be accompanied by a sharp and predicable rise in their variability, and thus a growing probability of extreme disturbance pulses (see Supplementary Fig. 1).

Here, we hypothesize that temporal dynamics of forest disturbances follow Taylor's law, such that the temporal variance in disturbance rate ($\mathrm{var}_d$) scales with the mean rate of disturbance ($\bar{x}_d$) as: $\mathrm{var}_d \propto \bar{x}_d{}^b$, where $\mathrm{var}_d$ and $\bar{x}_d$ are expressed as the percentage of total forest area that is disturbed per year, and $b$ is the power law exponent that describes how the variance changes with the mean. To test this hypothesis against data, we use high-resolution (30 m) annual maps of forest disturbance going back 35 years (1986–2020) and covering

[1]Technical University of Munich, School of Life Sciences, Freising, Germany. [2]Berchtesgaden National Park, Berchtesgaden, Germany. [3]University of Bristol, School of Biological Sciences, Bristol, UK. ✉e-mail: cornelius.senf@tum.de

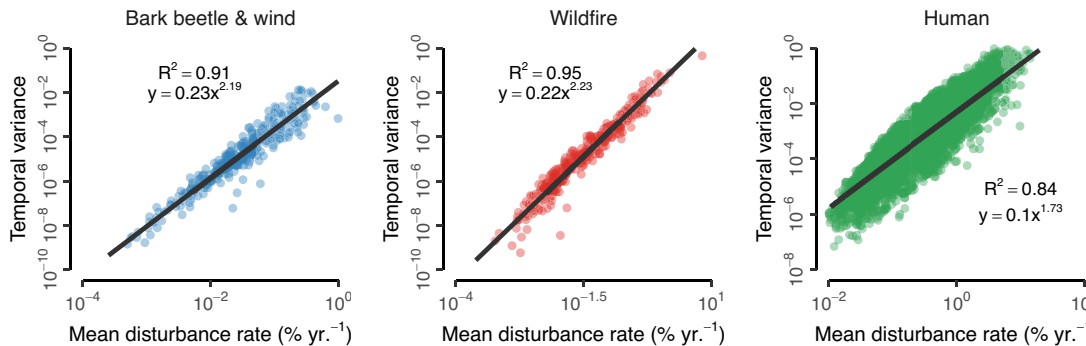

**Fig. 1 | Taylor's Law applies to temporal disturbance dynamics.** Shown are power law relationships between mean disturbance rate and temporal variance in disturbances rate for natural (bark beetles & wind [blue], wildfire [red]) and human disturbance agents (green). Note that both axes are log-scaled, and scaling is different for each plot.

more than 160 million ha of forests in Europe[27]. We calculate the temporal mean and variance of forest disturbance rates at various spatial grains (100–25,600 km²) and test if they follow a consistent power law relationship across scales, different natural disturbance agents (wildfire, bark beetle and wind disturbances; Supplementary Figs. 2, 3) and biomes (boreal, temperate, and Mediterranean; Supplementary Fig. 4). We also test if a similar power law relationship emerges for human disturbances associated with timber harvesting, which is the single most important driver of forest canopy openings in Europe and beyond[28,29]. We then compare our empirical results to simulated and analytical solutions for Taylor's law[26], assessing whether a power law scaling between mean disturbance rates and their variability over time is purely an outcome of statistical sampling, or the result of underlying ecological processes. Finally, we use a simulation approach to highlight the implications of a power law relationship between mean disturbance rates and their temporal variability for changing disturbance regimes.

## Results

We found a positive power law relationship between the mean and variance of annual forest disturbance rates (Fig. 1). Observed power law exponents were considerably outside the range of those expected under a random process of no relationship between mean and variance ($p < 0.001$; Supplementary Fig. 5). Mean disturbance rates and their temporal variability were tightly coupled, with model fits measured in terms of $R^2$ varying between 0.56 and 0.95, depending on the disturbance agent, spatial resolution and biome considered (Supplementary Figs. 6, 7). We thus found strong support for our hypothesis that forest disturbance dynamics can be described through Taylor's law. We chose the best fitting spatial grain for all subsequent analyses, which was 25,600 km² for all natural disturbances and 100 km² for human disturbances. The power law exponent was 2.19 (2.11–2.27; 95% confidence interval) for bark beetle and wind disturbances, and 2.23 (2.17–2.29) for wildfires (Fig. 1). The exponents of natural disturbances did not vary substantially between biomes and spatial grains (Supplementary Figs. 8, 9), suggesting a consistent power law scaling irrespective of disturbance agent, spatial scale or region.

Compared to natural disturbances, the power law exponent for human disturbance was considerably lower (1.73 [1.73–1.74]; Fig. 1). Overall, the power law exponent for human disturbances remained stable across spatial resolutions (Supplementary Fig. 8), but the relationship between mean and variance was strongest at smaller spatial grains (Supplementary Fig. 6). It also varied significantly between biomes, with a smaller exponent for boreal forests (1.44 [1.44–1.42]) compared to temperate and Mediterranean forests (1.69 [1.68–1.70] and 1.94 [1.93–1.95], respectively; Supplementary Fig. 9). So, while we

found that Taylor's law was able to describe temporal patterns of human disturbances just as well as those driven by natural agents, increases in mean harvest rates were associated with a less pronounced amplification of the temporal variability of human disturbance compared to natural disturbances, especially in temperate and boreal forests.

We compared our power law exponent estimates to a common statistical solution of Taylor's law (Fig. 2), which assumes the power law to result purely from identically and independently distributed sampling from a skewed distribution. Specifically, we compared our estimates to an analytical solution of this statistical artifact developed by Cohen and Xu[26], as well as to independently and identically distributed draws from a log-normal distribution calibrated with empirical means and variances. We found substantially lower power law estimates in our analysis compared to both the analytical and simulation-based statistical solution, which suggests that the observed pattern cannot be explained solely by identically and independently distributed sampling from a skewed distribution. For bark beetle and wind disturbances, both the simulated and analytical solutions suggest a power law coefficient of ~3–4, which is substantially larger than the exponent we observed empirically (~2.2). A similar difference was found for wildfires, where the simulated and analytical solution suggest exponents of ~2.5–4. For human disturbances, the simulated and analytical solution show less agreement, with predicted exponents ranging between 2 and 4.5, but still considerably larger than what was observed empirically (~1.7). Overall, these results suggest that scaling relationships between mean disturbance rates and their temporal variability identified in this study do not arise purely from statistical sampling but are likely also the result of additional underlying ecological mechanisms.

The power law relationships captured by our data have important implications considering recently observed increases in forest disturbance regimes globally (Fig. 3). Statistical simulations based on observed mean–variance scaling relationships for natural disturbances (power law exponent = 2.2) illustrate how increasing mean disturbance rates cause annual disturbance rates to quickly become more variable, and thus more likely to include extremely large disturbance pulses. For example, if we assume a long-term mean disturbance rate of 0.5% per year—which is consistent with observed rates for Europe in the late 20th century[30]—it is highly unlikely that annual disturbances rates would exceed 2.5% even in the most extreme years (probability <0.001 %; Fig. 3b). Doubling the mean disturbance rate to 1% yr.⁻¹—as has already occurred across Europe in the beginning of the 21st century[30] – increases the probability of experiencing a year with >2.5% annual disturbance rate to 1.2% (or once every 82 years). If mean disturbance rates were to rise to 2% yr.⁻¹ – which is possible based on latest projections[13,31]—we would expect a year with >2.5% annual

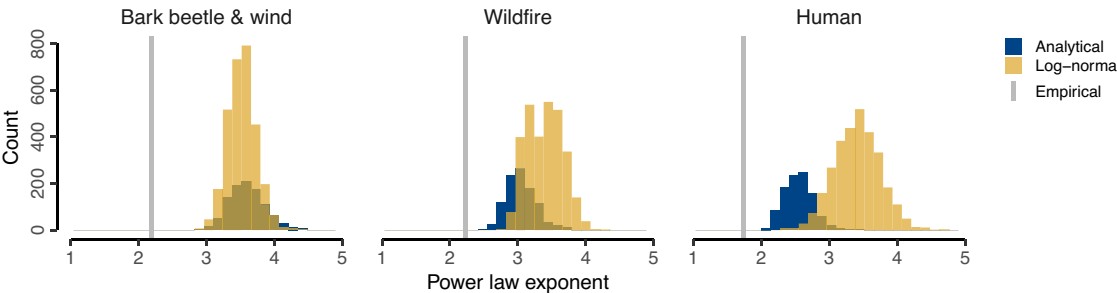

**Fig. 2 | Comparison of power law estimates to a common solution of Taylor's Law.** We compared our power law exponent estimates to a common statistical solution of Taylor's law (Fig. 2), which assumes the power law to result purely from identically and independently distributed sampling from a skewed distribution. Specifically, we compared our estimates to an analytical solution of this statistical artifact developed by Cohen and Xu[26], as well as to independently and identically distributed draws from a log-normal distribution calibrated with empirical means and variances. We found substantially lower power law estimates in our analysis compared to both the analytical and simulation-based statistical solution, which suggests that the observed pattern cannot be explained solely by identically and independently distributed sampling from a skewed distribution.

a)

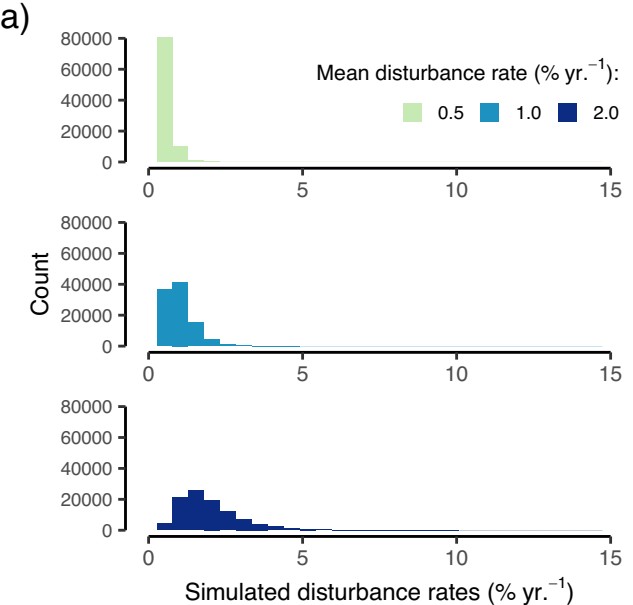

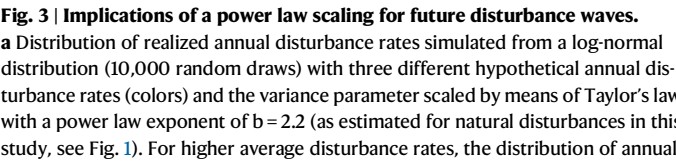

b)

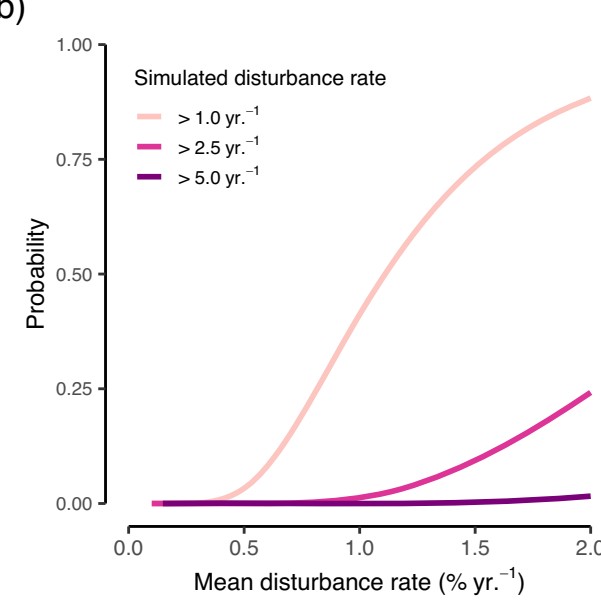

**Fig. 3 | Implications of a power law scaling for future disturbance waves.** **a** Distribution of realized annual disturbance rates simulated from a log-normal distribution (10,000 random draws) with three different hypothetical annual disturbance rates (colors) and the variance parameter scaled by means of Taylor's law with a power law exponent of b = 2.2 (as estimated for natural disturbances in this study, see Fig. 1). For higher average disturbance rates, the distribution of annual disturbance rates does not only shift but also increase in widths, leading to a higher probability of high disturbance rates in relation to the mean. **b** Changes in the probability of observing a year with annual disturbance rates greater than 1.0%, 2.5%, or 5.0% as the mean disturbance rate increases from 0 to 2% yr.$^{-1}$, assuming mean disturbance rate and temporal variability scale according to Taylor's law (power law exponent of b = 2.2).

disturbance rate to occur every four years (probability of 24 %) (Fig. 3b). Similarly, an extreme disturbance pulse with >5% annual disturbance rate, as observed in Central Europe in recent years[13], is highly unlikely under average disturbance rates <1% yr.$^{-1}$ (probability <0.001 %; Fig. 3b) but would be expected every 62 years (probability of 1.6 %) under average disturbance rates of 2 % yr.$^{-1}$.

## Discussion

We report a strong and consistent power law relationships between mean disturbance rates and their temporal variation for natural disturbances across Europe's forests, with a consistent power law exponent of ~2.2 across natural disturbance agents (wildfires, wind, and bark beetle), spatial grains, and biomes. The fact that a consistent power law relationship emerged irrespective of disturbance agent, spatial grain, or biome provides evidence that the temporal dynamics of natural disturbances can generally be described through Taylor's law. This finding bears significant implications for global forests.

Increases in mean disturbance rates have been observed in many forests worldwide[8] and our analysis shows that even modest changes in mean disturbance rates substantially increase temporal variance and thus the likelihood of years with extreme disturbance rates that far exceed historical averages. Events like the massive dieback of spruce in Central Europe since 2018[13,31,32] or extreme fires across Australia in 2019[33] were thus not unexpected or unpredictable. The notion of unpredictability of extreme events has been challenged earlier[34] and our results suggest that the prediction of future disturbance pulses could in fact be possible based on observations of mean disturbance rates alone. Given that mean disturbance rates are expected to further increase globally[18,19], it is highly likely that extreme disturbance pulses similar to those observed in recent years will occur more frequently in the future. Moreover, because of the heavy-tailed nature of disturbance rates, our models suggest that if mean disturbance rates continue to rise, in the future we may witness disturbance events that dwarf anything seen in recent history. Such "Dragon Kings"[35] often

indicate a bifurcation or tipping point, which might be identified using the power law relationships quantified here.

Many statistical and ecological explanations have been proposed to explain Taylor's law and some of those explanation might also apply to the power law scaling of temporal disturbance dynamics observed in our study. First and foremost, the power law identified in this study could be a statistical pattern emerging from the underlying distributional properties of disturbance rates, which are often right-skewed with heavy tails[36]. Sampling from skewed distributions has been shown to lead to power law scaling between mean and variance, with higher skewness leading to larger power law exponents[26]. The high skewness in the distribution of disturbance rates can be explained by their intrinsic relationship to climate variability[37,38]. Many meteorological variables linked to forest disturbances show skewed distributions themselves (e.g., gust windspeeds[39], heatwaves[40] or drought[41]), and mean-variance scaling has also been used to describe meteorological events (e.g., tornados[42], rainfall[43], or heatwaves[44]). The general power law scaling we identified for natural disturbances might thus be driven by the distributional properties of underlying climate events, with rare but large climate events causing rare but large disturbance pulses.

While the general power law scaling identified in our study likely emerges from the skewed distribution of underlying drivers, empirical power law exponents were smaller than those simulated from a lognormal distribution and exponents based on an analytical solution[26]. This discrepancy suggests that additional ecological mechanisms play a role in shaping the power law scaling of disturbance dynamics, dampening disturbance variability compared to independent and identical samples from a skewed distribution. Reasons for this dampening might be feedback between disturbance and vegetation. For example, once a large disturbance has hit an area, a subsequent large disturbance is less likely in the immediate future due to the self-limiting properties of most major natural disturbance agents (e.g., fire consuming fuel and insects killing their host trees)[45]. Likewise, frequent disturbances can lead to increased structural variability in forest canopies, dampening the spread of further disturbances[45], or resulting in less disturbance-prone species gaining competitive advantage[46]. The negative feedback between vegetation and disturbance thus likely results in lower power law exponents than expected based on purely statistical grounds.

We also found consistent power law scaling between mean disturbance rates associated with harvesting and their temporal variability, though with a lower power law exponent. Canopy openings from harvesting are thus less variable than natural disturbances in Europe, which might indicate that management successfully dampens the variation in forest clearings compared to purely natural processes at similar disturbance rates, especially at large spatial grain. The stronger power law scaling at smaller grain suggests, however, that harvest rates and their fluctuations over time can be highly dependent on localized forest landowners, which are heterogenous[47] and respond differently to markets and policies. Explanations for this response diversity are, for example, variable risk and time preferences, and different objectives (e.g., profit maximization versus heritage or conservation values). Finally, power-law behavior of mean and variance of human disturbances with a low scaling coefficient can also be reproduced mechanistically by behaviorally consistent models (e.g., considering response diversity of land managers in a simulation of human deforestation rates[48]). A power-law relationship between mean and variance for human disturbances is thus plausible, especially at small spatial grains.

We found significant variation in the power law exponent of human disturbances between biomes, some of which might be explained by interactions between human and natural disturbance regimes[29]. While boreal forests in Europe are intensively managed, they have been less affected by natural disturbances in the past[7], allowing for a steady management of timber resources (with reduced variance

at high mean harvest rates). Conversely, the pulses of natural disturbances affecting Central Europe in the past also strongly impacted the timber-based economy of the region (causing considerable market fluctuations) and limited the available resources for regular forest operations (e.g., harvesting equipment being sequestered in salvage logging operations). This, in turn, likely increased the temporal variance in human disturbance activities. Forest management and land use legacies vary considerably across Mediterranean forests, ranging from intensively managed plantations to forests recovering following land abandonment[49]. Mediterranean forests are also frequently exposed to climate extremes such as drought and heatwaves[50] and disturbances like wildfires, both of which influence management activities across the region[51]. Timber supply in Mediterranean systems is therefore likely to be more volatile over time than in boreal and temperate systems, ultimately leading to similar temporal variability to that associated with natural disturbances.

Our findings of disproportionate increases in extreme disturbance pulses with increasing mean disturbance rates have important consequences for managed forest systems. It is extreme disturbances that are a particular challenge to human wellbeing: Timber-based forest economics, for instance, suffers disproportionally from extreme disturbance events[52]. Furthermore, natural hazards such as flooding and debris flow increase disproportionally at high disturbance rates[53]. More broadly, an increasing likelihood of extreme events might overwhelm the capacity of society to respond to disturbances (e.g., by exceeding the funds available to respond to and provide relief from extreme disturbance pulses)[54]. Also, large pulses of disturbances can homogenize forest structure and demography, and ultimately result in more open, younger, and shorter forests in the future[55]. Forest management should react to the increasing likelihood of large disturbance pulses by accounting for disturbances in forest planning (e.g., by considering extreme disturbance pulses in infrastructure planning of road networks, timber storage facilities), but also through compensating for increasing disturbances in planned harvests[29]. Finally, forest policy should focus on fostering resilience to extreme disturbance pulses, as the ongoing changes in forests dynamics could push forests beyond their safe operating space.

## Methods

We used an existing, annual-resolution disturbance map of Europe[27] to derive disturbance rates, here defined as the annual percentage of forest area disturbed. The map was created from Landsat satellite data and depicts where and when a high severity disturbance has occurred between 1986 and 2020 at 30 m spatial resolution. Additionally, we used a satellite-based attribution product described in ref. 7 and recently extended until 2020 to attribute each disturbance to either harvest or natural causes[29], including one class for bark beetle and wind disturbance and one class for wildfire. Wind and bark beetle were combined in one class due to technical difficulties in separating these disturbance agents from the satellite data. While the map excludes other natural causes of disturbances (e.g., avalanches, flooding), they only account for a small proportion of natural disturbances in Europe and we thus assume them to be of negligible importance at a continental scale. Annual disturbance maps were aggregated to a coarser grid of variable spatial grain, starting with $10 \times 10$ km grid cells (100 km$^2$) and then doubling the cell width until reaching a grid cell size of $160 \times 160$ km (25,600 km$^2$). For each grid cell we first summed the total area disturbed per year and then divided it by the total forest area within the same grid cell to yield annual disturbance rates (expressed as a percentage). Finally, for each grid cell we calculated the mean ($\bar{x}_d$) and variance (var$_d$) of disturbance rates over the 35 years of our dataset (1986 to 2020).

We modeled the relationship between $\bar{x}_d$ and var$_d$ as a power law: var$_d = a * \bar{x}_d{}^b$, where $b$ is the exponent that describes how the variance changes with the mean and $a$ is a constant. $b = 1$ corresponds to a linear

relationship between $\bar{x}_d$ and $var_d$, while $b > 1$ indicates a disproportioned increase in $var_d$ with increasing $\bar{x}_d$. Parameters were estimated using Ordinary Least Squares (OLS) regression by fitting the following linear model to log-log transformed data: $\log(var_d) = a + b * \log(\bar{x}_d)$. We also tested alternative functional relationships (linear [$var_d = a + b*\bar{x}_d$], exponential [$\log(var_d) = a + b*\bar{x}_d$]), but found the Power law model to be most consistent with the data visually and in terms of $R^2$ (Supplementary Fig. 10). We fit separate models for wind and bark beetle disturbances (combined), wildfires and harvest to test for differences in the power law exponent between different natural and human disturbance agents. To test for significance of the power law exponent and model, we randomly shuffled the data and re-fitted the model. The reshuffling represents the null hypothesis of no relationship between mean and variance, with variability arising solely from sampling variability. Comparing our empirically estimated exponent to the distribution of exponents derived from these permutations allowed us to calculate the probability that the estimated exponents originate from the null model. We further compared the stability of the power law exponent across spatial resolutions (cells between 100 and 25,600 km$^2$) and biomes (boreal, temperate, and Mediterranean) by fitting individual models.

We compared our empirical estimates to theoretical estimates based on independent and identical distributed (*iid*) sampling from a skewed distribution (log-normal distribution), as well as to an analytical solution[26]. We used a log-normal distribution because disturbance rates are bounded to positive values and are often right skewed. The log-normal distribution was fitted to the data using the empirical mean ($\bar{x}$) and standard deviation ($sd$) following $X = \sigma Y + \mu$ with $Y \sim N(0, 1)$, $\mu = \log(\bar{x}) - 0.5 * \log((sd/\bar{x})^2 + 1)$ and $\sigma = \sqrt{\log((sd/\bar{x})^2 + 1)}$. Taking the exponent of $X$ yields disturbance rates on the original scale. We tested the assumption of a log-normal distribution for modeling observed disturbance rates by simulation analysis, comparing 10,000 random draws from the log-normal distributions calibrated with empirical means and variances against observed data (Supplementary Fig. 11). We also tested an alternative formulation using a squared normal distribution but found this did not represent bark beetle/wind and fire disturbances as well (Supplementary Fig. 11). The analytical solution is derived from dividing the empirical skewness by the empirical coefficient of variation and should approximate the expected power law exponent under *iid* sampling. To account for uncertainties, we bootstrapped both the sampling from the log-normal distribution and the analytical solution with 1000 repetitions each. We finally compared the empirical power law exponent to the distributions of expected power law exponents. If the empirical exponent falls outside the distribution of expected values, we consider this evidence that the observed power law scaling does not purely emerge from statistical sampling.

Finally, we derived 10,000 random draws from a log-normal distribution over a range of different mean disturbance rates ($\bar{x}_d = 0.5\%$, 1% and 2% yr$^{-1}$, representing historic, recent, and potential future disturbance rates in Europe[13,30,56]) and corresponding variances derived from the fitted power law relationship for natural disturbances. From the 10,000 simulations we calculated the probability of a year having an annual disturbance rate greater than 1.0%, 2.5% or 5.0%. The first threshold presents an average disturbance year of the last century[30]. The second threshold presents a historic extreme such as large-scale winter storms observed in the late 20th century[56]. The thirds threshold corresponds to recent extremes observed in Central Europe in response to the 2018/2019 drought[13]. All analysis were performed in R[57].

### Reporting summary
Further information on research design is available in the Nature Portfolio Reporting Summary linked to this article.

## Data availability
The disturbance maps used in this study are available from https://doi.org/10.5281/zenodo.3924380 (https://zenodo.org/records/7080016). All data, including already aggregated intermediate data products, are available from https://doi.org/10.5281/zenodo.15631399 (https://doi.org/10.5281/zenodo.15631399).

## Code availability
All code is available from following repository: https://github.com/corneliussenf/taylorslaw. A permanent version stored is stored on Zenodo at https://doi.org/10.5281/zenodo.15631399 (https://zenodo.org/records/15631400).

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

## Acknowledgements

R.S. acknowledges funding from the European Union's Horizon 2020 research and innovation program under grant agreement no. 101000574 (RESONATE: Resilient forest value chains enhancing resilience through natural and socio-economic responses) and from the European Research Council under the European Union's Horizon 2020 research and innovation program (Grant Agreement 101001905, FORWARD). T.J. was supported by a UK NERC Independent Research Fellowship (grant code: NE/SO1537X/1).

## Author contributions
C.S. developed the research idea and design, with input from T.J. C.S. performed all analysis and display items. R.S. and T.K. helped in interpreting the findings. All authors contributed to writing the manuscript.

## Funding

## Competing interests
The authors declare no competing interests.
