## [Transparent Peer Review file · Nature Communications]

Taylor's law predicts unprecedented pulses of forest disturbance under global change

Corresponding Author: Professor Cornelius Senf

Version 0:

Reviewer comments:

Reviewer #1

(Remarks to the Author)
Review for NCOMMS-24-09560-T

Taylor's law (TL) is a wide-spread statistical pattern and has been confirmed for numerous quantities. Motivated by the need to homogenize the variance for the analysis of variance procedure, TL was first tested for species abundance over space and time, but has been applied to other areas beyond ecology.

This submission aims to test the temporal TL for the forest disturbance rate in Europe, at different spatial scales, disturbance types, and biomes. The authors found that TL holds under different contexts, but shows different slopes, especially between natural and human disturbance types. They provided explanations for the observed form and the different parameters of TL. They used a simulation to predict how the observed mean-variance scaling can lead to more frequent extreme disturbance pulse under higher mean disturbance rate, thus offering insights into using TL to guide forest management and policy making.

This submission is well structured and clearly written. The arguments are well thought-out and the figures greatly support the empirical findings. I only have several comments for the authors to consider.

1. First, I would suggest the author to replace "coefficient" by "slope" or "exponent" in the text. This is because b , when studied through the power law on arithmetic scale, is actually the power exponent; when studied using the linear form on log-log scale, b becomes the slope of the linear equation. Using the word "coefficient" would easily be confused with the actual coefficient a in the original TL $\text{var} = a(\text{mean})^b$. It is important to choose the correct terminology so the results are not mis-interpreted.
2. In the extended figure 7, one can see that, at large spatial scales in the Mediterranean biome, the Taylor's law coefficient (actually the slope) for human disturbance is very similar to that for natural disturbance. This does not support the argument that "management successfully dampens the variation in forest canopies compared to purely natural disturbance regimes..." (lines 166-170). It will be interesting to know why in the Mediterranean, there is an elevated TL slope for human disturbance. Is it because of the heterogeneous landscape or the different forest management policy or some other reasons?
3. I wonder why the authors decide to combine the bark beetle and wind disturbance when analyzing TL. These two specific disturbances seem very different with regard to their types and impacts to the forest. I would suggest the authors to explain their decision about this data analysis procedure.
4. The paper does not define anywhere when TL is supported or contradicted. Should the definition be based on the coefficient of determination or the significance of TL's slope? If the latter, at what significance level? Adding these statistical details will make the statistical analysis more rigorous.
5. Given the intrinsic relation between natural disturbance and climate events, I suggest the authors to look into a couple of references that study TL for climate variables (Tippett and Cohen 2016; Tippett et al. 2020). I hope by reading these references it will inspire the authors to draw interesting connections between the two sets of studies. If not, at least it will provide the readers some context about the background of TL for non-biological quantities.

6. My last point is more of a big picture question about TL, which may not be the focus of this paper and does not have to be necessarily addressed by the authors. I list it here merely to share some background information about TL with the authors. One main question about TL is whether it is a pure statistical pattern or has biological meaning. This question has been debated in some early references of TL (Downing 1986; Taylor et al. 1988) and comes up again in recent years (Cohen and Xu 2015; Xiao et al. 2015). I am curious about what the authors think about this question in terms of their data. Does the observed TL reflect true forest disturbance dynamics or only a result of the spatial heterogeneity of the forest distribution? I am wondering if some kind of permutation test by randomly arranging the pair of disturbance rate and forest spatial distribution can demonstrate this.

References

- Cohen, J. E., & Xu, M. (2015). Random sampling of skewed distributions implies Taylor's power law of fluctuation scaling. *Proceedings of the National Academy of Sciences*, 112(25), 7749-7754.
- Downing, J. A. (1986). Spatial heterogeneity: evolved behaviour or mathematical artefact?. *Nature*, 323(6085), 255-257.
- Taylor, L. R., Perry, J. N., Woiwod, I. P., & Taylor, R. A. J. (1988). Specificity of the spatial power-law exponent in ecology and agriculture. *Nature*, 332(6166), 721-722.
- Tippett, M. K., & Cohen, J. E. (2016). Tornado outbreak variability follows Taylor's power law of fluctuation scaling and increases dramatically with severity. *Nature Communications*, 7(1), 10668.
- Tippett, M. K., & Cohen, J. E. (2020). Seasonality of Taylor's law of fluctuation scaling in all-India daily rainfall. *npj Climate and Atmospheric Science*, 3(1), 3.
- Xiao, X., Locey, K. J., & White, E. P. (2015). A process-independent explanation for the general form of Taylor's law. *The American Naturalist*, 186(2), E51-E60.

Reviewer #2

(Remarks to the Author)

The paper by Senf et al is conceptually interesting as it's generally informative when conceptual models can scale across organisms and systems. However, I have a number of concerns. In general, the mechanism underlying their results is not clearly presented. The methods are unclear, in particular the selection of spatial sizes, the time period of averaging, and also the omission of drought. The interpretations are unclear, which is probably why they seemed illogical at times. The figures were hard to decipher due to a lack of clear text and labels. Ultimately, it's nice when a general model can predict things, but without clear methods and importantly, without clear underlying mechanism, the value is greatly diminished. Also, is this just a statistical result? It makes sense that two variables will correlate when they are autocorrelated, so maybe this is just spurious? SPecific comments related to these points are detailed below.

Line 62, here it says 'the mean'. Is this the mean rate of disturbance? If so, I suggest clarifying as "the mean rate of disturbance".

Line 257, over what time period did you calculate the mean and variance of disturbance rates? Since rates are per unit time, this infers that the mean and variance were calculated over multiple years.

Line 66, why across scales? Maybe clarify why different scales were used. What is the benefit that this provides to our interpretation?

Line 68, what about drought?

Line 84, is there a difference between spatial scale and spatial grain? Maybe it would be best to use one word or the other to avoid confusion. Also, can you clarify why the scale varies with disturbance type? I can imagine lots of natural disturbances can be well below 25,600 km² in size. I would think they are quite variable in size. And I find it hard to imagine that harvests are 100km², particularly in Europe. That is huge. I didn't see an explanation of this in the methods. Maybe this is fine, but without understanding this rationale for static sizes, I'm left feeling quite concerned about how your results are influenced by these static sizes.

Line 92, frequency of extremes is not something you have mentioned in the paper yet. Is 'variance' the frequency of highs and lows, or is it the magnitude of highs and lows? How do we know that variance increases extremes? This needs explanation here.

Lines 118-121, the writing is unclear. For a 0.5% rate, you don't say the likelihood of a 5% event, but for 1.0% in the next sentence, you say it could occur once in 112 years. So what were the years for the 0.5% rate? 224?

Figure 2, I don't understand this figure. The legend does not adequately explain it. What is the difference between mean and realized disturbance rates? What parameter density is shown on the y-axis? For panel b, what is the difference between the mean and annual disturbance rates? How is "extreme" defined?

Lines 144-145, how is 'larger pulses' defined? Does this mean the area of mortality? If so, how did you show this in the results? It seemed you selected the best fit static areas for your analysis. I don't understand how your methods allow you to say the word 'larger'.

Lines 145-149, I would think that the reduction in likelihood of a disturbance following a previous one would result in reduced variance. Can you clarify the logic behind this sentence?

Lines 153-155, this could be logically backwards. Sure, areas that have more drought are exposed to more drought, but they also have trees adapted to drought. How do you account for this? Is there evidence to support your statement?

How can harvests fit Taylor's law? This suggests that the mean rate of harvests influences the mean variability. I don't see the mechanistic logic in that. Please clarify.

Extended figure 1 is unclear. What are the y and x axes? Without that, I am struggling to understand panel d (the lower left hand figure). By the way, labeling the panels a-d would help.

Extended figure 4, coefficient of determination of what? Some figure legends are quite vague, leaving the reader wondering what they are showing. Same comment for Extended figure 5.

Version 1:

Reviewer comments:

Reviewer #1

(Remarks to the Author)

The authors have addressed all my comments adequately and made the corresponding changes in the new manuscript. I have no other new comments to add.

Reviewer #3

(Remarks to the Author)

I don't think I'm an appropriate reviewer for this ms. I was expecting to be assessing whether the findings made sense in terms of ecological disturbance processes, and was not expecting to be bogged down in methodological and scaling questions. But the findings of this ms. seem so highly dependent on all of the somewhat opaque methodological scaling and aggregating considerations (that are beyond my experience and expertise to properly assess); thus (similar to original Reviewer 2) I can't get a solid grip on whether there is something actually going on of ecological or management interest in a real-world ecological-process way — or whether these Taylor's Law (TL)-looking relationships are primarily just outcomes of inherent properties of these datasets (e.g., of the right-tail skewness of underlying climate driver distributions) and/or the methods.

Thus I suggest that what you really need is another ecological scaling expert to assess this document.

Reviewer #4

(Remarks to the Author)

The authors sufficiently responded to most of the original comments from Reviewer 2, but I had some questions on the following original comments and responses:

"Comment 8" – this was originally regarding Fig 2 (renumbered to Fig 3). The original reviewer didn't understand the figure and I think it could be improved. For panel a) I found it confusing to have "realized" and "simulated" in the same description. Could other terms be used? If not, then maybe clarify more in the caption. For panel b) the legend is a bit unclear as presented. Are the different curves really rates "greater" (than) or are they equal to 1, 2.5, and 5? If so, remove "greater:" and if not, consider using ">" or at least add "than." Should the x axis be "Mean annual disturbance rate"? Caption L 168-169 "increase in widths" – edit.

"Comment 13" – Extended Fig 1: it's still not clear what the bottom panels represent.

In the new Fig 2, I find the text, "analytical" and "analytical solution" unclear, especial in contrast with the "log-normal." Edit and/or describe further in the caption.

I also noted many inconsistencies on how Taylor's law was spelled throughout the paper and supplemental material, which was surprising, given the title.

Reviewer #5

(Remarks to the Author)

I think the authors dealt with several of the comments by Reviewer #2 adequately. They do not however address much in terms of the comments about the lack of mechanism and questioning the importance of these patterns biologically.

Reviewer 2: "Ultimately, it's nice when a general model can predict things, but without clear methods and importantly,

without clear underlying mechanism, the value is greatly diminished. Also, is this just a statistical result? It makes sense that two variables will correlate when they are autocorrelated, so maybe this is just spurious?"

This comment was provided at the beginning of the review and does not have a specific response from the authors.

Reviewer #1 also addresses some of these points. Noting that there has long been a debate in the literature about exactly this - is Taylor's law interesting or just a statistical artefact?

I think, just because something is a statistical inevitability, does not mean it is not important. It is something that does seem likely to happen (to be specific, an increase in extremes of disturbance frequencies with increasing mean rates of disturbance).

The empirical tests of the pattern provide nice support that this is a real phenomenon that is likely occurring and will continue to occur.

Thus, on this point, I think the paper is still a valuable contribution.

However, the article would benefit from more explanation of what it means (and does not mean) in terms of ecological mechanisms for Taylor's law to explain the data well.

A few other points I noticed.

On

Line 20: "using Taylor's law, which relates the mean and variability of a system through a power law relationship."

This is not a good justification for the use of this model. Maybe rephrase to "which predict changes in variability and thus the frequency of extremes from changes in the mean."

Line 27: "The power law relationship was consistent" is misleading. It sounds like the parameters of the power law were the same across all of these things. Change to something like: "A power law was shown to be a good model for the patterns of xxx."

And in that vein, if a power law is the exact theoretical prediction of Taylor's law, then throughout you should provide statistical metrics to validate the likelihood that the observed data is governed by a power law. These are well-developed in Clauset et al. 2009, although there might be something more recent as well. In Clauset et al. they provide methods for finding a "p-value" a likelihood that your data could have been generated by your best-fit power law alone. I think they say values less than 10% are not good evidence.

The figure legend and text explaining Figure 2 are not sufficient. For example, in the legend, it does not state what the analytical solution plotted is for.

It seems like this analysis demonstrates that Taylor's law does not explain the data well. In particular, that reality is much less dramatic than Taylor's law would predict (lower power law exponent).

This is a big problem and requires a lot more care in the writing of the manuscript unless it is a misunderstanding of the analyses of Figure 2.

Reviewer #1:

Taylor's law (TL) is a wide-spread statistical pattern and has been confirmed for numerous quantities. Motivated by the need to homogenize the variance for the analysis of variance procedure, TL was first tested for species abundance over space and time, but has been applied to other areas beyond ecology.

This submission aims to test the temporal TL for the forest disturbance rate in Europe, at different spatial scales, disturbance types, and biomes. The authors found that TL holds under different contexts, but shows different slopes, especially between natural and human disturbance types. They provided explanations for the observed form and the different parameters of TL. They used a simulation to predict how the observed mean-variance scaling can lead to more frequent extreme disturbance pulse under higher mean disturbance rate, thus offering insights into using TL to guide forest management and policy making.

This submission is well structured and clearly written. The arguments are well thought-out and the figures greatly support the empirical findings. I only have several comments for the authors to consider.

Response: We thank the reviewer for the overall positive feedback and constructive suggestions, which helped us to greatly improve the manuscript.

Comment 1: First, I would suggest the author to replace "coefficient" by "slope" or "exponent" in the text. This is because b , when studied through the power law on arithmetic scale, is actually the power exponent; when studied using the linear form on log-log scale, b becomes the slope of the linear equation. Using the word "coefficient" would easily be confused with the actual coefficient a in the original TL $\text{var} = a(\text{mean})^b$. It is important to choose the correct terminology so the results are not mis-interpreted.

Response: We agree with the point raised by the reviewer and changed the word "coefficient" to "exponent" throughout the manuscript, which made the description of the results clearer.

Comment 2: In the extended figure 7, one can see that, at large spatial scales in the Mediterranean biome, the Taylor's law coefficient (actually the slope) for human disturbance is very similar to that for natural disturbance. This does not support the argument that "management successfully dampens the variation in forest canopies compared to purely natural disturbance regimes..." (lines 166-170). It will be interesting to know why in the Mediterranean, there is an elevated TL slope for human disturbance. Is it because of the heterogeneous landscape or the different forest management policy or some other reasons?

Response: We agree with the reviewer that our argumentation was not consistent and that we did not properly discuss the differences in power law exponents between boreal/temperate and Mediterranean forests. We revised the discussion to better reflect on potential reasons why Mediterranean forests behave different (L. 257):

"Forest management and land use legacies vary considerably across Mediterranean forests, ranging from intensively managed plantations to forests recovering following

land abandonment ¹¹. Mediterranean forests are also highly sensitive to climate extremes such as drought and heatwaves ² and disturbances like wildfires, both of which influence management activities across the region ³. Timber supply in Mediterranean systems is therefore likely to be more volatile over time than in boreal and temperate systems, ultimately leading to similar temporal variability to that associated with natural disturbances.”

Comment 3: I wonder why the authors decide to combine the bark beetle and wind disturbance when analyzing TL. These two specific disturbances seem very different with regard to their types and impacts to the forest. I would suggest the authors to explain their decision about this data analysis procedure.

Response: We thank the reviewer for this comment, as it highlighted a sub-optimal description of the dataset used in our study. We had to combine bark beetle and wind as in the original dataset both agents were combined due to methodological limitations with classification of the satellite imagery. We revised the methods description to clarify this important point:

“Additionally, we used a satellite-based attribution product described in ⁴ and recently extended until 2020 to attribute each disturbance to either harvest or natural causes, including one class for wildfire and one combined class for bark beetle and wind disturbances. Wind and bark beetle were combined in one class due to technical difficulties in separating these disturbance agents from the satellite data. While the map excludes other natural causes of disturbances (e.g., avalanches, flooding), they only account for a small proportion of natural disturbances in Europe and we thus assume them to be of negligible importance at a continental scale.”

Comment 4: The paper does not define anywhere when TL is supported or contradicted. Should the definition be based on the coefficient of determination or the significance of TL’s slope? If the latter, at what significance level? Adding these statistical details will make the statistical analysis more rigorous.

Response: We thank the reviewer for this comment. Indeed, we did not make clear on what statistic we base our main result. That said, we are critical of classical null hypothesis tests, i.e. assuming a zero slope as null hypothesis, which is likely to be rejected even if the data would not follow Taylor’s law. We hence decided for an alternative strategy, using a permutation test (i.e. randomly reshuffling data) to identify possible exponents under the null hypothesis (no relationship between mean and variance) and compare the distribution of these exponents to those estimated from our data. The results are shown in Extended Figure 5. We also note the permutation test results in the text (L.83):

“We found a positive power law relationship between the mean and variance of annual forest disturbance rates (Figure 1). Observed power law exponents were considerably outside the range of those expected under a random process of no relationship between mean and variance ($p < 0.001$; Extended Figure 5).”

Comment 5: Given the intrinsic relation between natural disturbance and climate events, I suggest the authors to look into a couple of references that study TL for

climate variables (Tippett and Cohen 2016; Tippett et al. 2020). I hope by reading these references it will inspire the authors to draw interesting connections between the two sets of studies. If not, at least it will provide the readers some context about the background of TL for non-biological quantities.

Response: We thank the reviewer very much for the suggested literature. We read the suggested papers with great interest. The power law identified in our study might indeed emerge from the distributional properties of underlying climate events, which themselves often can be characterized by Taylor's Law. To better reflect upon this important point, we substantially revised the discussion (L. 196; see also response to next comment):

"Many statistical and ecological explanations have been proposed to explain Taylor's law and some of those explanation might also apply to the power law scaling of temporal disturbance dynamics observed in our study. First and foremost, the power law identified in this study could be a statistical pattern emerging from the underlying distributional properties of disturbance rates, which are often right-skewed with heavy tails ⁵. Sampling from skewed distributions has been shown to lead to power law scaling between mean and variance, with higher skewness leading to larger power law exponents ⁶. The high skewness in the distribution of disturbance rates can be explained by their intrinsic relationship to climate variability ^{7,8}. Many meteorological variables linked to forest disturbances show skewed distributions themselves (e.g. gust windspeeds ⁹, heatwaves ¹⁰ or drought ¹¹) and mean-variance scaling has also been used to describe meteorological events (e.g., tornados ¹², rainfall ¹³ or heatwaves ¹⁴). The general power law scaling we identified for natural disturbances might thus be driven by the distributional properties of underlying climate events, with rare but large climate events causing rare but large disturbance pulses."

Comment 6: My last point is more of a big picture question about TL, which may not be the focus of this paper and does not have to be necessarily addressed by the authors. I list it here merely to share some background information about TL with the authors. One main question about TL is whether it is a pure statistical pattern or has biological meaning. This question has been debated in some early references of TL (Downing 1986; Taylor et al. 1988) and comes up again in recent years (Cohen and Xu 2015; Xiao et al. 2015). I am curious about what the authors think about this question in terms of their data. Does the observed TL reflect true forest disturbance dynamics or only a result of the spatial heterogeneity of the forest distribution? I am wondering if some kind of permutation test by randomly arranging the pair of disturbance rate and forest spatial distribution can demonstrate this.

Response: Again, we thank the reviewer for the great literature suggestions, which we were unaware of. After reading the paper by Cohen and Xu 2015 we decided to test our empirically estimated power law exponents against simulations from a skewed distribution as well as a corresponding analytical solution. If the power law identified in our study would emerge simply due to sampling variability, we would expect empirical results to match up with the simulations/analytical solution. However, we found that the empirical exponents are smaller than the expected exponents, suggesting that additional ecological mechanisms are needed to explain our empirical results. As we believe that this is an important finding, we incorporated these new results into our manuscript (new Figure 2) and revised the discussion accordingly (L. 215). We discuss which ecological mechanisms could cause the

smaller than expected exponents, that is dampen variability compared to simple random sampling from a skewed distribution. Again, we thank the reviewer for this great suggestion, which substantially improved our study.

Additional changes:

- We switched from a squared-normal to a log-normal for performing our simulations as we found better agreement of the log-normal with our data (see Extended Figure 10).
- We also fixed a small numerical error that occurred during creation of Figure 2 (now Figure 3). The error did not change the main message, though.
- We restructured the discussion to improve flow given our new results.
- To be consistent with previous studies, we used OLS for estimating the regression slope. This did not change the results.

Reviewer #2:

The paper by Senf et al is conceptually interesting as it's generally informative when conceptual models can scale across organisms and systems. However, I have a number of concerns. In general, the mechanism underlying their results is not clearly presented. The methods are unclear, in particular the selection of spatial sizes, the time period of averaging, and also the omission of drought. The interpretations are unclear, which is probably why they seemed illogical at times. The figures were hard to decipher due to a lack of clear text and labels. Ultimately, it's nice when a general model can predict things, but without clear methods and importantly, without clear underlying mechanism, the value is greatly diminished. Also, is this just a statistical result? It makes sense that two variables will correlate when they are autocorrelated, so maybe this is just spurious? Specific comments related to these points are detailed below.

Response: We thank the reviewer for the thorough review that helped us to improve the manuscript.

Comment 1: Line 62, here it says 'the mean'. Is this the mean rate of disturbance? If so, I suggest clarifying as "the mean rate of disturbance".

Response: Yes, this is the mean rate of disturbance. We have clarified this in manuscript, where we write: "[...] such that the temporal variance in disturbance rate (var_d) scales with the mean rate of disturbance (\bar{x}_d) as: $var_d \propto \bar{x}_d^b$, where var_d and \bar{x}_d are expressed as the percentage of total forest area that is disturbed per year, and b is the exponent that describes how the variance changes with the mean."

Comment 2: Line 257, over what time period did you calculate the mean and variance of disturbance rates? Since rates are per unit time, this infers that the mean and variance were calculated over multiple years.

Response: We calculate the mean and variance over a 35-year period and have clarified this in manuscript, where we write: "Finally, for each grid cell we calculated the mean (\bar{x}_d) and variance (var_d) of disturbance rates over the 35 years of our dataset (1986 to 2020)."

Comment 3: Line 66, why across scales? Maybe clarify why different scales were used. What is the benefit that this provides to our interpretation?

Response: We thank the reviewer for this comment. The mean and variance of disturbance rates can vary quite significantly across spatial scales. We were thus interested if the power law exponent would remain stable across spatial scales or whether the power law was scale dependent. We revised the sentence to clarify our rationale behind the analysis:

"We calculated the temporal mean and variance of forest disturbance rates at various spatial grains (100-25,600 km²) and tested if they follow a consistent power law relationship across scales, different natural disturbance agents (wildfire, bark beetle and wind disturbances; Extended Figure 2 and 3) and biomes (boreal, temperate, and Mediterranean; Extended Figure 4)."

Comment 4: Line 68, what about drought?

Response: Drought is an important factor in triggering disturbances (i.e. bark beetle outbreaks and fire disturbances), but drought itself rarely causes direct tree mortality severe enough to be considered as a disturbance in our analysis (i.e., complete canopy cover loss at a scale of at least 30*30m). While there have been occurrences of drought-related die-off in some regions in Europe ¹⁵, those events were rather localized and often difficult to map from satellites ¹⁶.

Comment 5: Line 84, is there a difference between spatial scale and spatial grain? Maybe it would be best to use one word or the other to avoid confusion. Also, can you clarify why the scale varies with disturbance type? I can imagine lots of natural disturbances can be well below 25,600 km² in size. I would think they are quite variable in size. And I find it hard to imagine that harvests are 100km², particularly in Europe. That is huge. I didn't see an explanation of this in the methods. Maybe this is fine, but without understanding this rationale for static sizes, I'm left feeling quite concerned about how your results are influenced by these static sizes.

Response: We thank the reviewer for this comment, which highlighted a sub-optimal description of our methodological approach. "Spatial scale" was used synonymously to "spatial grain" in our manuscript. To avoid confusion, we switched to "spatial grain" throughout the manuscript, as "spatial grain" is more widely understood in the ecological community ("spatial scale" is mostly used in the remote sensing/GIS community). Furthermore, we believe that there is a misunderstanding of how we aggregated disturbances in our study. To calculate annual disturbance rates, we needed a regular grid (of variable grain sizes) to be overlaid over the disturbance map (30 m spatial grain). In our manuscript, spatial grain refers to the size of this grid (i.e. the size of one grid cell). Within one grid cell, there can be a certain number of disturbance patches of certain sizes (from 0.18 ha [the minimum mapping unit of the map products, i.e. two 30 m pixels] to several thousands of ha [i.e. a large fire]). Most of the disturbance patches are small (<1ha ¹⁷). We calculated the sum of all disturbances per grid cell (i.e. what is the forest area disturbed per year per grid cell) and divided this by the forest area within the grid cell, yielding the annual disturbance rate. Spatial grain thus does not refer to the grain of disturbance patches themselves, but to the grain of the grid which we used for calculating annual disturbance rates (as shown in Extended Figure 2 and 3 in the revised manuscript). We revised the methods description to improve clarity:

"The annual disturbance maps were aggregated to a coarser grid of variable spatial grains, starting with a 10×10 km grid cell (100 km²) and then doubling this until we reached a grid cell size of 160×160 km (25,600 km²). For each grid cell we first summed the total area disturbed per year and then divided it by the total forest area within the same grid cell to yield annual disturbance rates (expressed as a percentage)."

Comment 6: Line 92, frequency of extremes is not something you have mentioned in the paper yet. Is 'variance' the frequency of highs and lows, or is it the magnitude of highs and lows? How do we know that variance increases extremes? This needs explanation here.

Response: We agree with the comment and removed the sentence from the results. We pick up the issue later in the discussion, where we clearly explain how changing variance will result in a higher likelihood of large disturbance pulses:

“Increases in mean disturbance rates have been observed in many forests worldwide¹⁸ and our analysis shows that even modest changes in mean disturbance rates substantially increase temporal variance and thus the likelihood of years with extreme disturbance rates (i.e., far off the average).”

Comment 7: Lines 118-121, the writing is unclear. For a 0.5% rate, you don't say the likelihood of a 5% event, but for 1.0% in the next sentence, you say it could occur once in 112 years. So what were the years for the 0.5% rate? 224?

Response: We agree with the reviewer that the writing was unclear. The likelihood for an average disturbance rate of 0.5 % yr.⁻¹ is close to zero, because of the power law scaling of variance described in our paper (and not halved, which would imply linear scaling). That is also the reason it jumps to almost 20 % when increasing annual disturbance rates to only 1.5 % yr.⁻¹. To improve clarity, we revised the text:

“For example, if we assume a long-term mean disturbance rate of 0.5 % per year – which is consistent with observed rates for Europe in the late 20th century¹⁹ – it is highly unlikely that annual disturbance rates would exceed 2.5 % even in the most extreme years (probability < 0.001 %; Figure 3 b). Doubling the mean disturbance rate to 1 % yr.⁻¹ – as has already occurred across Europe in the beginning of the 21st century¹⁹ – increases the probability of experiencing a year with > 2.5 % annual disturbance rate to 1.2 % (or once every 82 years). If mean disturbance rates were to rise to 2 % yr.⁻¹ – which is possible based on latest projections^{20,21} – we would expect a year with > 2.5 % annual disturbance rate to occur every four years (probability of 24 %) (Figure 3 b).”

Note: as we fixed a small numerical error in the calculations underlying Figure 2 (now Figure 3), the numbers have changed slightly. This does not affect the conclusions drawn from the figure.

Comment 8: Figure 2, I don't understand this figure. The legend does not adequately explain it. What is the difference between mean and realized disturbance rates? What parameter density is shown on the y-axis? For panel b, what is the difference between the mean and annual disturbance rates? How is “extreme” defined?

Response: We agree that the figure caption was not clear enough and we revised it accordingly (see text below). For (a) the data density is for the realized annual disturbance rates (as mentioned for the x-axis), which are random draws from a squared normal distribution (now updated to long-normal distribution) for the three different mean disturbance rates (colors) and assuming a Taylor law scaling between mean and variance (with $b = 2.2$ in the revised manuscript). The distribution thus shows hypothetical annual disturbance rates that could be observed under each mean disturbance rate. For (b) the difference between mean and annual is the same: the mean is the mean disturbance rate (as describe in the introduction and methods and highlighted again in response to Comment 2) and the annual disturbance rate is each annual realization (like annual mean temperature and long-term mean annual temperature). We agree that “extreme” was not well defined and we deleted it from

the figure legend and directly define “extreme” years by their corresponding annual disturbance rates.

Revised figure legend: “(a) Distribution of realized annual disturbance rates simulated from a log-normal distribution (10,000 random draws) with three different hypothetical annual disturbance rates (colors) and the variance parameter scaled by means of Taylors law with a power law exponent of $b = 2.2$ (as estimated for natural disturbances in this study, see Figure 1). For higher average disturbance rates, the distribution of annual disturbance rates does not only shift but also increase in widths, leading to a higher probability of high disturbance rates in relation to the mean. (b) Changes in the probability of observing a year with annual disturbance rates greater than 1.0 %, 2.5 % or 5.0 % as the mean disturbance rate increases from 0 % to 2 % yr.⁻¹, assuming a Taylor law scaling between mean disturbance rate and temporal variability (power law exponent of $b = 2.2$).”

Comment 9: Lines 144-145, how is ‘larger pulses’ defined? Does this mean the area of mortality? If so, how did you show this in the results? It seemed you selected the best fit static areas for your analysis. I don’t understand how your methods allow you to say the word ‘larger’.

Response: Large pulse here refers to a large area of disturbances within one year, as defined in the first paragraph of the introduction. We show this in the results in the simulation analysis (Figure 2, now Figure 3), where the probability of experiencing a year with high or very high disturbance rates compared to the mean (a disturbance pulse) increases exponentially with increasing mean disturbance rates. In the sentences referend to by the reviewer, we discuss potential mechanisms behind the power law scaling identified in our study.

Comment 10: Lines 145-149, I would think that the reduction in likelihood of a disturbance following a previous one would result in reduced variance. Can you clarify the logic behind this sentence?

Response: The paragraph has been revised due to feedback from Reviewer #1 and we now better explain the vegetation-disturbance feedback dampening variability compared to expected variability under random sampling from a skewed distribution.

Comment 11: Lines 153-155, this could be logically backwards. Sure, areas that have more drought are exposed to more drought, but they also have trees adapted to drought. How do you account for this? Is there evidence to support your statement?

Response: While the reviewer is certainly right with their statement, even in areas with trees adapted to drought there will be large-scale die-offs¹⁵. Our argument here is that the observed Power law relationship could be explained by exogenous processes such as climate extremes, which often also follow Taylors Law. For example, areas with frequent fire occurrence (because of their dry climate) are more likely to experience an extreme fire season (extremely dry years). Even though trees are adapted (and thus more resistant), those extreme years often overcome adaptive strategies as larger-scale drivers become more important (e.g. landscape connectivity, etc., see²²). The same effect applies to areas with frequent storms (more prone to wind extremes, e.g. western Europe) and biotic disturbances (more

prone to large-scale bark beetle outbreaks, e.g. as recently observed in Central Europe).

To better explain this point, and in response to comments from Reviewer #1, we completely revised the section as follows (L. 196):

“Many statistical and ecological explanations have been proposed to explain Taylor’s law and some of those explanation might also apply to the power law scaling of temporal disturbance dynamics observed in our study. First and foremost, the power law identified in this study could be a statistical pattern emerging from the underlying distributional properties of disturbance rates, which are often right-skewed with heavy tails ⁵. Sampling from skewed distributions has been shown to lead to power law scaling between mean and variance, with higher skewness leading to larger power law exponents ⁶. The high skewness in the distribution of disturbance rates can be explained by their intrinsic relationship to climate variability ^{7,8}. Many meteorological variables linked to forest disturbances show skewed distributions themselves (e.g. gust windspeeds , heatwaves ¹⁰ or drought ¹¹) and mean-variance scaling has also been used to describe meteorological events (e.g., tornados ¹², rainfall ¹³ or heatwaves ¹⁴). The general power law scaling we identified for natural disturbances might thus be driven by the distributional properties of underlying climate events, with rare but large climate events causing rare but large disturbance pulses.”

Comment 12: How can harvests fit Taylor’s law? This suggests that the mean rate of harvests influences the mean variability. I don’t see the mechanistic logic in that. Please clarify.

Response: While evident in the data, the power law scaling was less pronounced for human disturbances than for natural disturbances. We discuss reasons for a power law scaling of human disturbances in the manuscript (L. 232) but summarize them again here in the response: Human forest management is driven by external factors (e.g. market prices, labor availability, wood-processing industry) that can lead to temporal variability in harvest rates. We show that those temporal variations are larger when average harvest rates are higher, likely because unexpected events such as fluctuating wood prices or disruptions in the processing chain will have larger impacts. We also show that TL was most supported for small spatial grains, indicating that such fluctuations cancel out across larger regions (compensation effects). Finally, natural and human disturbances are not independent, with forest management responding to natural disturbances and the power law observed for human disturbances might thus also result from the temporal variability of natural disturbances, even though dampened (and thus with a lower power law exponent). To improve the clarity of our arguments, we revised the discussion in starting in L. 230.

Comment 13: Extended figure 1 is unclear. What are the y and x axes? Without that, I am struggling to understand panel d (the lower left hand figure). By the way, labeling the panels a-d would help.

Response: We revised the figure legend to improve clarity.

Extended figure 4, coefficient of determination of what? Some figure legends are

quite vague, leaving the reader wondering what they are showing. Same comment for Extended figure 5.

Response: We revised the figure legends to improve clarity.

Additional changes:

- We added a new analysis comparing our empirical results to simulation/analytical solutions of Taylor's Law, showing that our empirical results yield lower power law exponents than expected under random sampling from a skewed distribution. We show this in Figure 2 and in the results paragraph starting in L. 117.
- We switched from a squared-normal to a log-normal for performing our simulations as we found better agreement of the log-normal with our data (see Extended Figure 10).
- We also fixed a small numerical error that occurred during creation of Figure 2 (now Figure 3). The error did not change the main message, though.
- We restructured the discussion to improve flow given our new results.
- To be consistent with previous studies, we used OLS for estimating the regression slope. This did not change the results.

References

1. Scarascia-Mugnozza, G., Oswald, H., Piussi, P. & Radoglou, K. Forests of the Mediterranean region: gaps in knowledge and research needs. *For. Ecol. Manag.* **132**, 97–109 (2000).
2. Gentilesca, T., Camarero, J., Colangelo, M., Nolè, A. & Ripullone, F. Drought-induced oak decline in the western Mediterranean region: an overview on current evidences, mechanisms and management options to improve forest resilience. *IForest - Biogeosciences For.* **10**, 796–806 (2017).
3. Moreira, F. *et al.* Wildfire management in Mediterranean-type regions: paradigm change needed. *Environ. Res. Lett.* **15**, 011001 (2020).
4. Senf, C. & Seidl, R. Storm and fire disturbances in Europe: distribution and trends. *Glob. Change Biol.* **27**, 3605–3619 (2021).
5. Maroschek, M., Seidl, R., Poschlod, B. & Senf, C. Quantifying patch size distributions of forest disturbances in protected areas across the European Alps. *J. Biogeogr.* *n/a*, (2023).
6. Cohen, J. E. & Xu, M. Random sampling of skewed distributions implies Taylor's power law of fluctuation scaling. *Proc. Natl. Acad. Sci.* **112**, 7749–7754 (2015).
7. Seidl, R. *et al.* Globally consistent climate sensitivity of natural disturbances across boreal and temperate forest ecosystems. *Ecography* **43**, 967–978 (2020).
8. Sommerfeld, A. *et al.* Patterns and drivers of recent disturbances across the temperate forest biome. *Nat. Commun.* **9**, 4355 (2018).
9. Justus, C. G., Hargraves, W. R., Mikhail, A. & Graber, D. Methods for Estimating Wind Speed Frequency Distributions. *J. Appl. Meteorol. 1962-1982* **17**, 350–353 (1978).
10. Guirguis, K., Gershunov, A., Cayan, D. R. & Pierce, D. W. Heat wave probability in the changing climate of the Southwest US. *Clim. Dyn.* **50**, 3853–3864 (2018).
11. Stagge, J. H., Tallaksen, L. M., Gudmundsson, L., Van Loon, A. F. & Stahl, K. Candidate Distributions for Climatological Drought Indices (SPI and SPEI). *Int. J. Climatol.* **35**, 4027–4040 (2015).
12. Tippett, M. K. & Cohen, J. E. Tornado outbreak variability follows Taylor's power law of fluctuation scaling and increases dramatically with severity. *Nat. Commun.* **7**, 1–7 (2016).
13. Tippett, M. K. & Cohen, J. E. Seasonality of Taylor's law of fluctuation scaling in all-India daily rainfall. *Npj Clim. Atmospheric Sci.* **3**, 1–7 (2020).
14. Schär, C. *et al.* The role of increasing temperature variability in European summer heatwaves. *Nature* **427**, 332–336 (2004).
15. Hammond, W. M. *et al.* Global field observations of tree die-off reveal hotter-drought fingerprint for Earth's forests. *Nat. Commun.* **13**, 1761 (2022).
16. Hartmann, H., Adams, H. D., Anderegg, W. R. L., Jansen, S. & Zeppel, M. J. B. Research frontiers in drought-induced tree mortality: crossing scales and disciplines. *New Phytol.* **205**, 965–969 (2015).
17. Senf, C. & Seidl, R. Mapping the forest disturbance regimes of Europe. *Nat. Sustain.* **4**, 63–70 (2021).
18. Allen, C. D. *et al.* A global overview of drought and heat-induced tree mortality reveals emerging climate change risks for forests. *For. Ecol. Manag.* **259**, 660–684 (2010).
19. Senf, C., Seibald, J. & Seidl, R. Increasing canopy mortality impacts the future demographic structure of Europe's forests. *One Earth* **4**, 1–7 (2021).

20. Senf, C. & Seidl, R. Persistent impacts of the 2018 drought on forest disturbance regimes in Europe. *Biogeosciences* **2021**, 5223–5230 (2021).
21. Hermann, M. *et al.* Meteorological history of low-forest-greenness events in Europe in 2002–2022. *Biogeosciences* **20**, 1155–1180 (2023).
22. Raffa, K. F. *et al.* Cross-scale Drivers of Natural Disturbances Prone to Anthropogenic Amplification: The Dynamics of Bark Beetle Eruptions. *BioScience* **58**, 501–517 (2008).

Reviewer #1:

Comment 1: The authors have addressed all my comments adequately and made the corresponding changes in the new manuscript. I have no other new comments to add.

Response: We thank the reviewer for the positive feedback

Reviewer #3:

I don't think I'm an appropriate reviewer for this ms. I was expecting to be assessing whether the findings made sense in terms of ecological disturbance processes, and was not expecting to be bogged down in methodological and scaling questions. But the findings of this ms. seem so highly dependent on all of the somewhat opaque methodological scaling and aggregating considerations (that are beyond my experience and expertise to properly assess); thus (similar to original Reviewer 2) I can't get a solid grip on whether there is something actually going on of ecological or management interest in a real-world ecological-process way — or whether these Taylor's Law (TL)-looking relationships are primarily just outcomes of inherent properties of these datasets (e.g., of the right-tail skewness of underlying climate driver distributions) and/or the methods. Thus I suggest that what you really need is another ecological scaling expert to assess this document.

Response: We are grateful for this frank reviewer comment on the lack of fit between the reviewer and the topic.

Reviewer #4:

The authors sufficiently responded to most of the original comments from Reviewer 2, but I had some questions on the following original comments and responses:

Comment 1: “Comment 8” – this was originally regarding Fig 2 (renumbered to Fig 3). The original reviewer didn’t understand the figure and I think it could be improved. For panel a) I found it confusing to have “realized” and “simulated” in the same description. Could other terms be used? If not, then maybe clarify more in the caption. For panel b) the legend is a bit unclear as presented. Are the different curves really rates “greater” (than) or are they equal to 1, 2.5, and 5? If so, remove “greater:” and if not, consider using “>” or at least add “than.” Should the x axis be “Mean annual disturbance rate”? Caption L 168-169 “increase in widths” – edit.

Response: For (a) we believe that realized is the correct term, because a simulation from a probability distribution (log-normal in our case) is a *realization* from this distribution. We understand, however, that this term could be misleading, and we thus replaced it with “simulated disturbance rates”. For (b) it is indeed greater than the respective threshold, and we followed the reviewer’s suggestion and added a “>” in front of each threshold (plus a specific unit). We also agree that the x-axis label was not specific enough and we changed it to “Mean disturbance rate (% yr.⁻¹)”

Comment 2: “Comment 13” – Extended Fig 1: it’s still not clear what the bottom panels represent.

Response: We revised Extended Figure 1 to clarify what is shown in the lower panels.

Comment 3: In the new Fig 2, I find the text, “analytical” and “analytical solution” unclear, especial in contrast with the “log-normal.” Edit and/or describe further in the caption.

Response: We thank the reviewer for this comment, which highlighted a sup-optimal figure caption. The ambiguity of the figure caption was also brought up by another reviewer. We revised the figure caption and the text explaining Figure 2 to make the use of the analytical solution clearer.

Comment 4: I also noted many inconsistencies on how Taylor’s law was spelled throughout the paper and supplemental material, which was surprising, given the title.

Response: We checked the full manuscript and now use one spelling consistently.

Reviewer #5:

Comment 1: I think the authors dealt with several of the comments by Reviewer #2 adequately. They do not however address much in terms of the comments about the lack of mechanism and questioning the importance of these patterns biologically.

Reviewer 2: "Ultimately, it's nice when a general model can predict things, but without clear methods and importantly, without clear underlying mechanism, the value is greatly diminished. Also, is this just a statistical result? It makes sense that two variables will correlate when they are autocorrelated, so maybe this is just spurious?"

This comment was provided at the beginning of the review and does not have a specific response from the authors.

Reviewer #1 also addresses some of these points. Noting that there has long been a debate in the literature about exactly this - is Taylor's law interesting or just a statistical artefact?

I think, just because something is a statistical inevitability, does not mean it is not important. It is something that does seem likely to happen (to be specific, an increase in extremes of disturbance frequencies with increasing mean rates of disturbance). The empirical tests of the pattern provide nice support that this is a real phenomenon that is likely occurring and will continue to occur. Thus, on this point, I think the paper is still a valuable contribution.

However, the article would benefit from more explanation of what it means (and does not mean) in terms of ecological mechanisms for Taylor's law to explain the data well.

Response: We generally agree that it is of interest to understand why average disturbance rate and variability in disturbance rate are Power law distributed, and that a point of critique in the previous review round was a lack of ecological explanation. We addressed this issue in the first round of revision by a substantial revision of the discussion, providing several explanations (i.e. from a purely statistical perspective [L202ff] and from a more ecological perspective [L. 2018ff]). We thus believe that there is sufficient interpretation of the results in ecological context. Testing those hypothetical explanations is, however, beyond the scope of the manuscript and should be done in future research (as this would also require a more process-based approach, not an observational as our approach). We thus believe that there is already sufficient ecological explanation of what can cause the observed statistical pattern, and we refrained from adding further details.

With respect to the comment by Reviewer #2 that was not properly answered by us ("Ultimately, it's nice when a general model can predict things, but without clear methods and importantly, without clear underlying mechanism, the value is greatly diminished. Also, is this just a statistical result? It makes sense that two variables will correlate when they are autocorrelated, so maybe this is just spurious?") we would like to note that we answered this general comment in the several smaller comments of his/her review. We also note that there is no indication of autocorrelation in our study, and we thus cannot follow the reviewer's comment that autocorrelation leads to spurious correlation explaining the patterns we observe.

Comment 2: Line 20: "using Taylor's law, which relates the mean and variability of a system through a power law relationship." This is not a good justification for the use of this model. Maybe rephrase to "which predict changes in variability and thus the frequency of extremes from changes in the mean."

Response: Thanks for the suggestion, which we adopted.

Comment 3: Line 27: "The power law relationship was consistent" is misleading. It sounds like the parameters of the power law were the same across all of these things. Change to something like: "A power law was shown to be a good model for the patterns of xxx."

Response: We thank the reviewer for the suggestion. However, Extended Figures 8 and 9 show that the coefficient is indeed consistent across natural disturbance agents and biomes. There's only a difference for human disturbances. As the "consistent" refers to "natural disturbances", we strongly believe that the sentence is correct. We thus refrained from changing it.

Comment 4: And in that vein, if a power law is the exact theoretical prediction of Taylor's law, then throughout you should provide statistical metrics to validate the likelihood that the observed data is governed by a power law. These are well-developed in Clauset et al. 2009, although there might be something more recent as well. In Clauset et al. they provide methods for finding a "p-value" a likelihood that your data could have been generated by your best-fit power law alone. I think they say values less than 10% are not good evidence.

Response: We thank the reviewer for this comment, but we politely disagree with the critique. Taylor's law is defined as a *functional* power law relationship ($y = ax^k$) between mean and variance. That is, there is no distributional assumption behind Taylor's Law. Our manuscript consequently does not test whether a random variable follows a Power law distribution (what is done in the paper by Clauset et al. 2009), but whether there is a functional Power law relationship between the empirical mean and variance of a random variable. We test for the functional Power law relationship (i.e. for conformity with Taylor's Law) by plotting the empirical means and variances in log-log space ($\log(y) = \log(a) + k * \log(x)$; Figure 1), where we find clear indication of a linear relationship (and thus a power law relationship on the original data scale) with exponents >0 (based on our permutation test shown in Extended Figure 5). Our results thus demonstrate that the observed pattern is consistent with Taylor's Law (i.e. the relationship between mean and variance follow a functional Power law relationship). One could compare the Power law functional relationship against other functional relationships. If another functional relationship fits the data better, this could be evidence that Taylor's Law is not the best theoretical model. Alternative functional relationships could be a linear model in original scale ($y = a + k * x$) or an exponential model in original scale ($\log(y) = a + k * x$), which both – visually – clearly do not fit the data well (see Figure below). There is, of course, a distributional assumption in fitting the functional relationships, for which we use maximum likelihood of the normal distribution (which simplifies to Ordinary Least Squares in our case). This distributional assumption could be tested (though we see no indication of heteroskedasticity or any dependency pattern that might violate the maximum likelihood assumption), but this wouldn't affect the functional relationship but the

estimates of a and k . We added a sentence to the methods that we also tested alternative functional relationships (L. 315): “We also tested alternative functional relationships (linear [$var_d = a + b * \bar{x}_d$], exponential [$\log(var_d) = a + b * \bar{x}_d$]), but found the Power law model to be most consistent with the data visually and in terms of R^2 (not shown here).”

Figure 1: Alternative functional relationships between mean and variance.

Comment 5: The figure legend and text explaining Figure 2 are not sufficient. For example, in the legend, it does not state what the analytical solution plotted is for. It seems like this analysis demonstrates that Taylor's law does not explain the data well. In particular, that reality is much less dramatic than Taylor's law would predict (lower power law exponent). This is a big problem and requires a lot more care in the writing of the manuscript unless it is a misunderstanding of the analyses of Figure 2.

Response: While we agree that important information was lacking in the figure caption, we disagree that the figures proves that the relationship between variance and mean does not follow Taylor's Law. The figure is about the *strength* of the Power law relationship, not about *if* a Power law relationship is present. That is, Power laws (or Taylor's Law if applied to variance~mean scaling) can have different exponents (strength of scaling) while still being Power law distributed. Like a simple linear model remains still a linear model if the regression slope varies. The figure simply shows

that we would expect a stronger Power law scaling under random sampling from a log-normal distribution and from a purely analytical solution. That is, while the relationship between mean disturbance rate and temporal variance follows Taylor Law, the strength of relationship is weaker than expected from a purely statistical perspective. This shows that there have to be additional processes at work explaining our empirical results *beyond* a simple statistical relationship between mean and variance (what we call dampening). We discuss this in detail in L 216ff. We hence politely disagree with the critique of the reviewer, as we firmly believe that his/her point of critique ("It seems like this analysis demonstrates that Taylor's law does not explain the data well.") is technically wrong. We revised the figure caption and text describing Figure 2, though, making our result clearer and less ambiguous.